# Fabrication of Erbium-Doped Upconversion Nanoparticles and Carbon Quantum Dots for Efficient Perovskite Solar Cells

**DOI:** 10.3390/molecules29112556

**Published:** 2024-05-29

**Authors:** Alhanouf Alotaibi, Farah Alsardi, Fatimah Alshwikhat, Madawey Aldossary, Fudhyah S. Almarwani, Faizah J. Talidi, Shouq A. Almenhali, Sarah F. Almotawa, Yahya A. Alzahrani, Sultan Alenzi, Anwar Alanazi, Masfer Alkahtani

**Affiliations:** 1Department of Physics, College of Science, Imam Abdulrahman Bin Faisal University, P.O. Box 1982, Dammam 3144, Saudi Arabia; 2200006057@iau.edu.sa (A.A.); 2190002998@iau.edu.sa (F.A.);; 2Future Energy Technologies Institute, King Abdulaziz City for Science and Technology (KACST), P.O. Box 6086, Riyadh 11442, Saudi Arabiaaqalanazi@kacst.edu.sa (A.A.)

**Keywords:** solar energy, perovskite solar cell, upconversion nanoparticles, lithium, CQDs, photovoltaic performance

## Abstract

Upconversion nanoparticles (UCNPs) and carbon quantum dots (CQDs) have emerged as promising candidates for enhancing both the stability and efficiency of perovskite solar cells (PSCs). Their rising prominence is attributed to their dual capabilities: they effectively passivate the surfaces of perovskite-sensitive materials while simultaneously serving as efficient spectrum converters for sunlight. In this work, we synthesized UCNPs doped with erbium ions as down/upconverting ions for ultraviolet (UV) and near-infrared (NIR) light harvesting. Various percentages of the synthesized UCNPs were integrated into the mesoporous layers of PSCs. The best photovoltaic performance was achieved by a PSC device with 30% UCNPs doped in the mesoporous layer, with PCE = 16.22% and a fill factor (FF) of 74%. In addition, the champion PSCs doped with 30% UCNPs were then passivated with carbon quantum dots at different spin coating speeds to improve their photovoltaic performance. When compared to the pristine PSCs, a fabricated PSC device with 30% UCNPs passivated with CQDs at a spin coating speed of 3000 rpm showed improved power conversion efficiency (PCE), from 16.65% to 18.15%; a higher photocurrent, from 20.44 mA/cm^2^ to 22.25 mA/cm^2^; and a superior fill factor (FF) of 76%. Furthermore, the PSCs integrated with UCNPs and CQDs showed better stability than the pristine devices. These findings clear the way for the development of effective PSCs for use in renewable energy applications.

## 1. Introduction

Solar power is solar energy that has been captured, transformed, and used for electrical and thermal energy. The two major ways that solar energy is used are photovoltaic and photothermal. One of these, photovoltaic, directly generates electricity from solar energy using solar cells [1,2]. Crystalline silicon photovoltaics account for 95% of the market share and are the most widely used variety. Mono-Si and Multi-Si exhibit minimal toxicity and have profound industrial applications that were built along with the development of the electronic industry, while c-Si solar cells have a high operating efficiency of about 26.7% [3]. The chemistry required to purify, reduce, and crystallize pure silicon from sand, which is extremely energy-demanding and environmentally damaging, is the only thing holding crystalline Si back from becoming the perfect PV material [4]. GaAs solar cells, which were developed from second-generation solar cells, are an extremely efficient technology. Their efficiency has exceeded 30%, but they are far too costly for use in large-area terrestrial applications [5].

The cost reduction of modules from first- to second-generation solar cells is a positive move. Still, prices have not been sufficiently affordable for major commercial acceptance by clients due to the inherent difficulties of such devices. Due to their cost-competitiveness and streamlined manufacturing processes, third-generation solar cells have gained increased attention. Among the top contenders, with a significant increase in efficiency, are the recently researched perovskite materials, which appear to be appealing alternatives because to their relatively low prices and high efficiency [6]. These perovskites’ important characteristics are their simplicity of manufacturing, high solar absorption, and low nonradiative carrier recombination rates for such easily synthesized materials [7,8]. Because of the exceptional properties of perovskite materials, solar cells made from perovskite (PSCs) have achieved spectacular, unparalleled breakthroughs, achieving over 26% power conversion efficiency in the last ten years [9].

Organic–inorganic halide perovskites, with the chemical formula ABX_3_, consist of Cs or organic ammonium ions (A), such as Formamidine or Methylamine (CH_3_NH_3_^+^ or NH = CHNH_3_^+^); divalent metal cations (B), like Pb^2^^+^ or Sn^2^^+^; and halide ions (X), including I⁻, Br⁻, or Cl⁻. The combination of the A-site Cs or organic ammonium ions and X-site halogens forms the foundation of the most stable and influential halide perovskite solar absorbers, resulting in high power conversion efficiency (PCE) due to a suitable bandgap for significant photo-harvesting capabilities, extended charge carrier lifespan (τ), and high mobility (μ) [6,7,9,10,11]. Perovskite solar cells are exceptional in terms of enhanced performance; yet, they have several disadvantages that have delayed their marketing because of their sensitivity to oxygen, moisture, temperature, and UV light. Due to PSCs’ narrow absorption cross-section, only the visible spectrum of the sun can be efficiently absorbed, which reduces the efficiency of the solar cells [6].

Studies have been conducted to increase the PCE and stability of PSCs, such as improving their photovoltaic efficiency by either improving the perovskite active layer itself by advanced techniques such as laser-assisted crystallization [12,13] or adding additional light-gathering materials to PSCs in order to make the most of the available sunlight [14,15]. For example, upconversion nanoparticles (UCNPs) are one of the most straightforward ways to include NIR active elements in PSCs. Because of their capacity to create a single high-energy photon from two or more low-energy photons, upconversion nanoparticles (UCNPs) are a possible choice to utilize NIR. When exposed to NIR, erbium (Er^3+^)-doped UCNPs release light at green and red wavelengths that are successfully absorbed by the photoactive perovskite material [16,17,18]. Furthermore, upconversion nanoparticles (UCNPs) operating as scattering points can expand the path of light and encourage the development of larger, less imperfect perovskite grains, which is helpful for improving photovoltaic performance [15,19]. Additionally, several studies have been carried out to dope the electron/hole transport layer with a variety of substances in order to enhance its electrical properties [20,21,22]. Giordano et al. demonstrated that Li doping speeds up electron transport inside mesoporous TiO_2_ electrodes and showed that PSCs constructed on such electrodes exhibit superior performances compared to undoped electrodes, increasing PCEs from 17.0% to 19.3% [22].

Recent studies have indicated that passivating the grain boundaries of the perovskite layer with CQDs leads to notable enhancements in PSC efficiency and environmental stability. Studies have shown a remarkable increase in the photon conversion efficiency of PSCs when incorporating CQDs into the appropriate layers of the devices. For instance, the efficiency of PSCs based on CQD-modified perovskite films improved from 17.59% to 19.38% [23,24,25]. This enhancement is attributed, in part, to the hydrophobic nature of CQD molecules, which effectively shield the perovskite layer from moisture intrusion. Remarkably, even after four months of storage in a non-humidity-controlled environment, a CQD-modified perovskite retained its original black hue, underscoring its exceptional long-term stability [23].

In this study, we introduce a straightforward and highly efficient method for synthesizing lithium fluoride-based crystals doped with lanthanide ions, specifically LiYF_4_, and seamlessly integrating them into the mesoporous layers of perovskite solar cells (PSCs). To examine the impact of incorporating upconversion nanoparticles (UCNPs) into PSCs, we assembled fully functional PSC devices and meticulously evaluated their performance. The photovoltaic performance of PSCs incorporating the synthesized UCNPs exhibited an excellent improvement. Compared to control PSCs without UCNPs, they had a notable increase in power conversion efficiency (PCE) to 16.22% and a fill factor (FF) of 74%.

Furthermore, we explored the optimization of PSCs doped with 30% UCNPs by passivating them with carbon quantum dots (CQDs) at varying spin coating speeds. This optimization strategy led to significant enhancements in photovoltaic performance. For instance, compared to pristine PSCs, a fabricated device incorporating 30% UCNPs that was passivated with CQDs at a spin coating speed of 3000 rpm demonstrated an improved PCE, from 16.65% to 18.15%; a higher photocurrent, from 20.44 mA/cm^2^ to 22.25 mA/cm^2^; and an elevated fill factor (FF) of 76%. These results underscore the potential of a straightforward and effective approach utilizing UCNPs to augment the photovoltaic performance of PSCs, promising substantial advancements in renewable energy technologies.

## 2. Results and Discussion

Experimentally, UCNPs with core and core–shell structures were hydrothermally synthesized following a synthesis procedure previously reported in [26] and detailed in the Materials and Methods. To visualize the size and shape of the produced core and core–shell UCNPs, a few drops of each sample were placed on a transmission electron microscope (TEM) with a carbon-coated copper grid. Figure 1a shows well-dispersed core UCNPs with an average size of 18 nm. The particle size of the synthesized core UCNPs was then confirmed to be 18 nm using a dynamic light scattering machine (DLS), as illustrated in Figure 1b. Figure 1c shows the uniform structure of the synthesized UCNP core–shell nanocrystals, with an average size of 25 nm. The average size of the core–shell UCNPs was also confirmed with DLS to be 25 nm, as shown in Figure 1d.

The primary objective of employing core–shell-structured upconversion nanoparticles (UCNPs) in this research was to mitigate surface quenching effects. These effects arose from the interaction between the solvent ligands, specifically hydroxyl (OH) groups, and the upconverting ion, erbium (Er^3+^), used in this study. This interaction led to an undesirable energy transfer from the intermediate state of Er^3+^ to the solvent ligands, which could significantly dampen luminescence efficiency [26,27,28]. By strategically designing an inert shell around the core of the UCNPs, it was anticipated that there would be a substantial enhancement of visible upconversion (UC) luminescence when these nanoparticles were excited by infrared light. This enhancement was crucial for improving the optical performance of the perovskite solar cells. The addition of the inert shell effectively isolated the upconverting ions from the external environment, thereby maximizing the emission intensity and stability of the UCNPs.

Next, the synthesized core–shell UCNPs were incorporated into the mesoporous layers of the perovskite solar cells (PSCs) using various mixing ratios with titanium dioxide (TiO_2_), as illustrated in Figure 2a. This step was crucial for converting the near-infrared (NIR) solar spectrum into visible light, which the perovskite active layer was capable of absorbing. Achieving an optimal alignment between the emission spectrum of the upconverting rare-earth ion and the light-harvesting absorption band of the perovskite was essential for this process. Erbium (Er^3+^), in particular, has shown considerable promise due to its ability to emit intense red and green light, as recorded from the synthesized core-only UCNPs (18 nm). These emissions effectively matched the visual absorption spectrum of the perovskite active layer, enhancing the overall light absorption capabilities of the PSCs, as depicted in Figure 2b. This alignment was pivotal for maximizing the efficiency of the light conversion within the solar cells [29].

To optically investigate the upconverted light emission from the UCNPs and the emission of the perovskite light-harvesting layer, we fabricated a PSC device without gold electrodes to allow for light transmission. In this device, UCNPs with a core–shell structure with an average size of 25 nm were mixed with TiO_2_ nanoparticles in a mesoporous layer at a mixing ratio of 30%:70% (volume ratio), as this mixing ratio was found to give the highest PSC photovoltaic performance, which will be discussed later in this study. To optically characterize the fabricated layers, we designed and built a custom-made confocal laser-scanning microscope equipped with continuous-wave (CW) 532 nm (green) and 980 nm (NIR) lasers, an optical spectrometer, and a single-photon counter, as illustrated in Figure 3a. The PSC device was placed on the optical setup and irradiated with the 980 nm laser on both sides. In the case of Figure 3b, the optical emission of the synthesized core–shell UCNPs (with an average size of 25 nm) was collected by the optical spectrometer, which consisted of two bands in the green region and one band in the red region. Optical luminescence in UCNPs is a consequence of several consecutive transfers of energy between the activator (Er^3+^) and sensitizer (Yb^3+^). In the beginning, Yb^3+^ absorbs the first NIR photon and excites it to its ^2^F_5/2_ excited state, since at 950–1000 nm, Yb^3+^ exhibits a significant absorption cross-section, and afterwards it uses the energy from Yb^3+^ (in the excited state) to push Er^3+^ up to the semi-resonant metastable ^4^I_11/2_ level. A second NIR photon from Yb^3+^ boosts Er^3+^ to higher excited states because the semi-resonant metastable (^4^I_11/2_) level in Er^3+^ has a long millisecond life. Moving from the ^2^H_11/2_, ^4^S_3/2_, and ^4^F_9/2_ excited states of Er^3+^ to the ^4^I_15/2_ ground state generates two strong and sharp emission lines, with corresponding emission peaks at 527 nm, 553 nm, and 650–680 nm, after several nonradiative relaxations [29,30].

The optical emission from the perovskite material was investigated under green (532 nm) excitation. The photoluminescence of the perovskite film peaked at 780 nm with 30% doped UCNPs within the mesoporous layer, as shown in Figure 3c. Optical emission at 780 nm from perovskite materials occurs through a series of processes starting with the absorption of photons, which excite electrons from the valence band to the conduction band, creating excitons. These excitons undergo radiative recombination, releasing photons if the material’s band gap aligns with the energy corresponding to a wavelength of 780 nm. By tailoring the composition of the perovskite, such as adjusting the halide components or metal cations, the band gap can be optimized to specifically enhance emission at this near-infrared wavelength. This capability allows for precise control over the emission properties, making perovskites highly suitable for applications requiring emissions at specific wavelengths, like optical devices and advanced solar cells.

To investigate the photovoltaic performance of the PSCs with and without UCNPs in the core–shell structure (with an average size 25 nm), we manufactured several PSC devices with different ratios of UCNPs to TiO_2_ in the mesoporous layer. The fabricated PSC devices were named the pristine device, the device with 20% UCNPs, the device with 30% UCNPs, and the device with 50% UCNPs. To ensure reproducibility, we fabricated five devices under each set of conditions and calculated the average photovoltaic performance values. The photovoltaic measurements of the fabricated PSC devices were performed at 1.5 AM under one-sun illumination. The photovoltaic performances of the fabricated PSCs with and without UCNPs are summarized in Table 1 and shown in Figure 4.

The J-V results presented in Table 1 and illustrated in Figure 4a–c show intriguing trends in the performance metrics of the fabricated perovskite solar cells. Notably, the solar cell incorporating a 30% concentration of UCNPs exhibited the highest open-circuit voltage (Voc), FF, and PCE values among the devices tested. The enhancement of PCE was particularly remarkable, with a substantial increase of 5%. Furthermore, we also observed that there was a discernible correlation between the increase in the UCNP concentration up to 30% and the increases in FF and open-circuit voltage (Voc).

The improvement in performance observed for the device with 30% UCNPs can be attributed to the direct conversion of NIR light into additional photocurrent. This phenomenon was facilitated by the presence of integrated UCNPs within the mesoporous layer, effectively transforming a considerable portion of the NIR light into visible light that could be absorbed by the perovskite layer. Moreover, the fill factor experienced a notable increase from 71.6% to 74.3% as the UCNP concentration increased (0–30%). This improvement cannot be solely attributed to the light-harvesting ability of the UCNPs; rather, the lithium dopant present in the UCNP crystals also contributed to enhancing electron transport within the TiO_2_ layer. This reduction in deep traps enhanced the fill factor and open-circuit voltage. Furthermore, the enhanced performance of the perovskite solar cells can be attributed to the unique optical properties of the core–shell UCNPs. These nanoparticles exhibited superior scattering effects and specialized upconversion luminescence, which augmented the absorption of the perovskite. Additionally, they facilitated the production of larger perovskite grains with fewer imperfections, further enhancing device performance.

We found that the photovoltaic performance of the PSC device doped with 50% UCNPs significantly dropped, which could have been a consequence of excessive light back-scattering, which lowered absorption by reflecting a significant part of the incident light out of the solar cell. Furthermore, a larger UCNP injection resulted in a lack of conductivity in the electron transport layer, as demonstrated by the decreases in fill factor values in Table 1 and Figure 4. The results presented in Table 1 and Figure 4 are not the highest achieved in this study; however, they adequately demonstrate the optimal concentration of UCNPs for use in CQDs passivation, which will be further discussed in the next experiment.

Next, the champion PSCs doped with 30% UCNPs were then passivated with carbon quantum dots at different spin coating speeds to improve their photovoltaic performance, as illustrated in Figure 5a. For this, 100 μL of CQDs were added to the perovskite layer, and in the mesoporous layer the UCNP-to-TiO_2_ ratio of 30:70 was fixed. The thickness of the CQD layers was controlled via varying the revolutions per minute of the spin coating procedure. The devices were named as follows: PSC device 30% UCNPs, PSC device 30% UCNPs/3000 rpm CQDs, PSC device 30% UCNPs/5000 rpm CQDs, and PSC device 30% UCNPs/6000 rpm CQDs. To ensure reproducibility, we fabricated five devices under each condition and calculated the average photovoltaic performance values reported. The CQD layer deposited using spin coating was influenced by various factors such as the solution concentration, viscosity, spin time, and ambient conditions (temperature and humidity). To improve reproducibility, we conducted these experiments at different speeds and specified them. The non-electroactive passivation layer materials on the perovskite films were kept minimal to avoid blocking charge carriers. Typically, the passivation layer thickness (5–10 nm) is not visible by SEM and is not very easy to detect. It is therefore included within the error of repeatable experiments.

Table 2 and Figure 5b–d show the photovoltaic performances of the manufactured PSCs. The device named PSC device 30% UCNPs/3000 rpm CQDs showed the best overall performance and revealed the best photocurrent density (JSC) and PCE, with an increase in the Jsc value from 20.44 to 22.25 (mA/cm^2^) and a rise in PCE from 16.65% to 18.15%, which was a 9.009% increase when compared to the reference device (PSC device 30% UCNPs). The photovoltaic performances of the fabricated perovskite solar cells (PSCs) integrated with UCNPs and CQDs were significantly improved. This enhancement was primarily due to the ability of the UCNPs to convert low-energy incident photons into high-energy photons (UV and visible light). These high-energy photons were then absorbed and converted to electrons by the active layers of the PSCs, generating more electrons per photon compared to PSCs without UCNPs, thereby boosting the overall photovoltaic performance [15,19]. Additionally, incorporating CQDs contributed to the stability and efficiency of the PSCs by passivating the perovskite grain boundaries. The CQDs were also expected to broaden the absorption spectrum of the PSCs by converting high-energy photons, which could potentially damage the PSC structure, into low-energy photons that were more readily absorbed by the perovskite active layer. This resulted in the enhanced power conversion efficiency of the PSCs [24,25].

Figure 5c,d present the J-V characteristic curves of the PSC devices with different rotational speeds of CQD insertion. The PSC device 30% UCNPs/3000 rpm CQDs exhibited the best J-V characteristic curve, as it showed higher Jsc, FF, and maximum power values. This enhancement was caused by the increased number of photons that were absorbed by the perovskite layer. The Voc (open-circuit voltage) of PSC device 30% UCNPs/5000 rpm CQDs was found to be higher when passivated with carbon quantum dots (CQDs) at a spin coating speed of 5000 rpm. This improvement can be attributed to the enhanced surface passivation provided by the CQDs, resulting in fewer surface defects and reduced charge recombination. Additionally, the optimized optical properties and improved charge extraction efficiency achieved through CQD passivation contributed to the higher Voc. The stability of the perovskite layer also may have been enhanced at this passivation speed, further supporting the observed increase in Voc.

Furthermore, the fill factor (FF) of the PSC device with 30% UCNPs and CQDs spin-coated at 3000 rpm exhibited a peak value of 76%, surpassing that of the pristine device (Figure 5d). This outcome suggests that optimal doping of UCNPs within the mesoporous layer, coupled with the addition of 100 µL of CQDs at a spin coating speed of 3000 rpm, facilitated enhanced electron extraction from the perovskite film. Consequently, this improved the overall conductivity of the electron transport layer, leading to significant increases in both the power conversion efficiency (PCE) and fill factor (FF) values.

This study underscores the enhanced performance of perovskite solar cells (PSCs) through the integration of UCNPs and CQDs. Control devices without any doping served as a baseline, achieving a power conversion efficiency (PCE) of 15.5%. Incorporating UCNPs alone led to notable improvements, with the 30% UCNP device demonstrating the highest efficiency at 16.22%, which was attributed to improved light absorption and electron transport. Comparatively, devices with only CQDs, especially those processed at a 3000 rpm spin coating speed, also showed significant gains, achieving a PCE of 16.80%. This highlights the effectiveness of CQDs in enhancing photovoltaic performance by passivating perovskite grain boundaries and improving charge extraction.

When examining the combined usage of UCNPs and CQDs, the results were even more promising. The device with 30% UCNPs and CQDs added at 3000 rpm exhibited the highest performance, with a Jsc of 22.25 mA/cm^2^, an FF of 76.0%, a Voc of 1.075 V, and a PCE of 18.15%, representing substantial improvements over both the pristine device and those with individual doping. This combination outperformed the 30% UCNP device (PCE of 16.22%) and the 3000 rpm CQD device (PCE of 16.80%), demonstrating the synergistic effects of integrating both additives. The comparative analysis revealed that while the UCNPs enhanced the PSCs’ performance by converting near-infrared light to visible light and improving electron transport [15], the CQDs contributed by passivating defects and expanding light absorption [24,25]. The optimal doping of 30% UCNPs combined with 3000 rpm CQDs maximized these benefits, leading to the highest overall efficiency. This study highlights the potential to significantly improve PSC performance through strategic material integration, surpassing the gains achievable using UCNPs or CQDs alone [15,24,25].

The reported efficiency improvement in the perovskite solar cells (PSCs) from 16.22% to 18.15% is notable, representing an 11.9% relative increase. However, this should be benchmarked against the current highest laboratory efficiencies, which have reached around 25.7% [31] under standard testing conditions (1000 W/m^2^ irradiance, 25 °C, and AM 1.5 G spectrum) for small-area cells. While these record efficiencies showcase the potential of PSCs, achieving similar performance in larger, commercially viable modules that maintain stability over time remains a significant challenge. The ongoing research is essential to bridge the efficiency gap and enhance the scalability and real-world applicability of PSC technology.

Finally, to assess the stability of both UCNP- and CQD-modified perovskite solar cells (PSCs) under conditions with unregulated humidity, a thorough investigation was conducted over several days, as shown in Figure 6. This study compared the stability of three types of fabricated devices: PSC devices containing 30% UCNPs and 3000 rpm CQDs, PSC devices with 30% UCNPs only, and pristine devices. Performance was monitored over time, revealing notable differences. Remarkably, devices integrated with both UCNPs and CQDs exhibited superior stability compared to their pristine counterparts. Specifically, the PCE of the PSC devices with 30% UCNPs and CQD passivation at 3000 rpm maintained 92.5% of its initial value, while the pristine devices saw a decrease to 66% efficiency. This enhanced stability is attributed to the UCNPs’ ability to convert near-infrared light to visible light, improving electron transport, and the CQDs’ role in passivating grain boundaries, reducing defect densities in the perovskite film [15,24,25]. Additionally, CQDs interacted with uncoordinated lead ions in grain boundaries, mitigating moisture-induced degradation.

## 3. Materials and Methods

### 3.1. Nanoparticles: Core Preparation

In a two-neck flask, a mixture of 1.0 mmol of LnCl_3_ (Ln = Y (80.0 wt.%), Yb (18.0 wt.%), and Er (2.0 wt.%)), 10.5 mL of 1-octadecene, and 10.5 mL of oleic acid was heated to 150 °C for 40 min at atmospheric pressure under argon flow until it became a clear yellow solution. The solution was brought down to 50 °C. Then, 2.5 mmol of LiOH.H_2_O with 5.0 mL of methanol and 4.0 mmol of NH_4_F with 10.0 mL of methanol were mixed and gradually introduced. The mixture was then vigorously stirred for 40 min while the temperature was kept at 50 °C. To remove the methanol and residual water, we raised the temperature to 150 °C for 20 min. The generated LiYF_4_: Yb,Er UCNPs were then gathered, rinsed three times with ethanol, and reconfigured in 10 mL of chloroform after the solution had cooled to room temperature.

### 3.2. Nanoparticles: Core–Shell Preparation

In a two-neck flask, a mixture of 1.0 mmol of YCI_3_, 10.5 mL of 1- octadecene, and 10.5 mL of oleic acid was heated to 150 °C for 40 min at atmospheric pressure under argon flow until it became a clear yellow solution. The solution was brought down to 50 °C. Then, 2.5 mmol of LiOH.H_2_O with 5.0 mL of methanol, 4.0 mmol of NH_4_F with 10.0 mL of methanol, and 10 mL of an upconversion nanoparticle core solution were gradually introduced. The mixture was then vigorously stirred for 40 min while the temperature was kept at 50 °C. To remove the methanol and residual water, we raised the temperature to 150 °C for 20 min. The generated LiYF_4_: Yb,Er CS UCNPs were then gathered, rinsed three times with ethanol, and reconfigured in 10 mL of chloroform after the solution had cooled to room temperature.

### 3.3. Preparation of Ligand-Free UCNPs

We prepared acidic ethanol with a pH of 1 by adding 2.5 μL of hydrochloric acid to 40 mL of ethanol. Then, 1 mL of UCNPs were added to the acidic ethanol and sonicated for 1 h to remove the oleate ligands. After that, ligand-free UCNPs were collected using a centrifugation device and cleaned three times with ethanol. Then, in order to be used again, the oleate-free Ln-UCNPs were reconfigured in pure ethanol.

### 3.4. Carbon Quantum Dot-Based Glucose Preparation

Carbon quantum dots (CQDs) were derived from glucose by dissolving 2 g in 15 mL of distilled water, followed by the addition of 6 mL of a 25% aqueous ammonia solution. This mixture was subjected to hydrothermal synthesis at 180 °C under a pressure of 3 MPa for 1 h, ensuring optimal reaction conditions. After synthesis, the resulting brown suspension underwent purification using dialysis bags with a molecular weight cutoff of 3000 KDa for 12 h, effectively removing impurities. Subsequently, the purified solution was subjected to filtration to eliminate any remaining large particles. Low-speed centrifugation at 6000 rpm for 10 min concentrated the CQDs in the solution. Finally, the concentrated solution was carefully stored in isopropanol for further experimentation or application.

### 3.5. Preparation of Perovskite Solar Cell

The substrates used to fabricate the PSCs had dimensions of 1.6 cm × 2.45 cm (FTO glass (Fluorine-doped Tin Oxide was purchased from Sigma-Aldrich, St. Louis, MO, USA)). Using zinc powder and 4 molar hydrochloric acid (HCl) (Sigma-Aldrich), we etched an FTO layer 0.5 cm from the top side of the substrates to separate the cathode from the anode and create an open circuit. All cleaning was carried out using an ultrasonic bath. The glass substrates were cleaned by sonication in distilled water and Hellmanex soap (Ossila, Sheffield, UK) for 30 min, then distilled water only for 10 min, ethanol (Fisher, Waltham, MA, USA) for 15 min, and acetone for 10 min. Lastly, the substrates were dried with air to evaporate the acetone and placed in a UV-Ozon cleaner for 20 min.

### 3.6. Preparation of Compact Layer by Spray Pyrolysis

In a vial, we prepared a compact solution by adding 600 μL of Titanium diisopropoxide bis(acetylacetonate) (Sigma-Aldrich), 400 μL of Acetylacetone (Sigma-Aldrich), and 900 μL of Ethanol (Fisher). All of the FTO substrates were prepared by placing them on a hot plate at 450 °C for 30 min while covering the anode area. The thin compact TiO_2_ layer was sprayed 3 times using a process known as spray pyrolysis. After completing the spray pyrolysis, the substrates were maintained at 450 °C for 30 min.

### 3.7. Deposition of Mesoporous TiO_2_ and UCNP-Doped Mesoporous TiO_2_ Using Spin Coating

A mesoporous solution was prepared by mixing 30 NR-D titanium dioxide (TiO_2_) paste (Greatcell, Queanbeyan, Australia) and ethanol at a ratio of 1:6 (*v*/*v*). We worked on 4 concentrations of lithium-based UCNPs, and they were 0%, 20%, 30%, and 50% of the previously prepared UCNPs. Adhesive tape was placed on only one side (anode tip) to protect the anode area from the mesoporous TiO_2_ layer. The UCNP-doped mesoporous TiO_2_ was deposited by spin coating 50 μL of the different concentrations (program: 20 s with an acceleration of 2000 and a speed of 4000). The substrates were placed on the hot plate at 450 °C for 30 min.

### 3.8. Perovskite and Spiro-OMeTAD Layers for Samples without CQDs

Both the perovskite layer and the spiro layer were prepared inside a Nitrogen glove box. A perovskite solution was prepared by weighing 21.54 mg of Cesium Iodide (CsI) (abcr), 18.89 mg of Methylammonium Bromide (MABr) (Greatcell), 62.04 mg of Lead bromide (PbBr_2_) (Tokyo Chemical Industry, TCI, Tokyo, Japan), 247.2 mg of Formamidinium Iodide (FAI) (Greatcell), and 722.4 mg of Lead iodide (PbI_2_) (Alfa aesar, Ward Hill, MA, USA) dissolved with 960 μL of Dimethyl sulfoxide (DMSO) (ACROS, Antwerpen, Belgium) and 2400 μL of Dimethylformamide (DMF) (ACROS). The solution was mixed and heated to 90 °C to dissolve the materials, and 50 μL of the precursor solution was placed onto the glass/FTO/compact-TiO_2_/non-doped and UCNP-doped mesoporous TiO_2_ in two steps: a 10 s spin at 1000 rpm followed by a 30 s spin at 6000 rpm. To remove residual DMSO and DMF, 200 μL of chlorobenzene (ACROS) was applied to the spinning wet perovskite film for the last 18 s, and the films were then heated at 100 °C for 30 min on a hot plate. We prepared Spiro MeOTAD by adding 102.72 mg of spiro, 1200 μL of chlorobenzene, 21.36 μL of Lithium bis(trifluoromethane)sulfonimide (Sigma Aldrich) (where 520 mg of Li was dissolved in 1000 μL of acetonitrile (CAN) (ACROS), and 34.52 μL of 4-tert-Butylpyridine (TBP) (Sigma Aldrich). After that, 50 μL of a 2,2′,7,7′-Tetrakis[N,N-di(4-methoxyphenyl)aminuteso]-9,9′-spirobifluorene (spiro-OMeTAD) (Xi’an Yuri Solar Co., Ltd., Xi’an, China) solution was spin-coated at 4000 rpm for 20 s to create the hole transfer layer (HTL), which was placed on top of the perovskite layer. Finally, 80 nm of metal contact Au was evaporated onto the prepared perovskite films using thermal evaporation.

### 3.9. Preparing the Perovskite and Spiro-OMeTAD Layers with Doped CQDs for Samples with CQDs

A perovskite solution with a composition of Cs0.05MA0.10FA0.85Pb(Br0.10I0.85)3 was prepared inside a Nitrogen glove box along with spiro-OMeTAD. To prepare the perovskite solution, the following materials were dissolved: 18.89 mg of methylammonium bromide (MABr), 247.2 mg of formamidine Iodine (FAI), 722.4 mg of lead iodide (PbI2), 62.04 mg of lead bromide (PbBr2), and 21.54 mg of cesium iodide (CsI). Additionally, 960 µL of dimethyl sulfoxide (DMSO) and 2400 µL of dimethylformamide (DMF) were added. The solution was heated at 90 °C on a hot plate for 30 min. Subsequently, the solutions and substrates were transferred to a dry-air glove box with humidity lower than 2%. Using spin coating, 50 µL of the precursor solution was applied in a two-step method with a lower-RPM mode (acceleration: 200 RPM/s, velocity: 1000 RPM, time 10 s) followed by a higher-RPM mode (acceleration: 2000 RPM/s, velocity: 6000 RPM, time 30 s). At 18 s before the end of spinning, 200 µL of chlorobenzene was applied to the wet film to remove residual DMSO and DMF. The substrates were then heated to 100 °C for 45 min on a hot plate to create crystalline triple-cation perovskite layers [2,3]. After that, we deposited 100 μL of dispersed CQDs in Isopropanol above the perovskite films and put them on the middle of the substrates, which rotated at 3000, 5000, or 6000 rpm. The substrates were then taken and placed on the hot plate at 100 °C for 5 min. Next, in the Nitrogen glove box, we mixed 10.72 mg of spiro, 1200 µL of chlorobenzene, 21.36 µL of lithium, and 34.52 µL of 4-tetra-Butylpyridine to create spiro-OMeTAD. After that, the perovskite layer was covered with a hole transfer layer (HTL) by spin coating 50 µL of a spiro-OMeTAD solution for 20 s at 4000 rpm. Finally, a gold layer (80 nm) was deposited using thermal evaporation at a certain pressure to form metal contact electrodes.

## 4. Conclusions

In conclusion, this study focused on synthesizing upconversion nanoparticles (UCNPs) doped with erbium ions to facilitate the conversion of ultraviolet (UV) and near-infrared (NIR) light into usable energy. These UCNPs were successfully integrated into the mesoporous layers of perovskite solar cells (PSCs) at various concentrations. The highest photovoltaic performance was achieved by PSCs incorporating 30% UCNPs in their mesoporous layers, yielding a power conversion efficiency (PCE) of 16.22% and a fill factor (FF) of 74%. Subsequently, passivation of the champion PSCs, also doped with 30% UCNPs, with carbon quantum dots (CQDs) at different spin coating speeds further improved their photovoltaic performance. Specifically, the PSC device passivated with CQDs at 3000 rpm exhibited a significantly enhanced PCE of 18.15%, a photocurrent increased from 20.44 mA/cm^2^ to 22.25 mA/cm^2^, and a superior fill factor (FF) of 76% compared to the pristine PSCs. Furthermore, the PSCs integrated with UCNPs and CQDs showed better stability than the pristine devices. The reported results show the UCNPs’ ability to convert near-infrared light to visible light, improving electron transport, and the CQDs’ role in passivating grain boundaries, reducing defect densities in the perovskite film. Additionally, CQDs interacted with uncoordinated lead ions in grain boundaries, mitigating moisture-induced degradation. These findings demonstrate that UCNPs effectively convert near-infrared light to visible light, enhancing electron transport. Simultaneously, CQDs play a crucial role in passivating grain boundaries, thereby reducing defect densities in perovskite film. Moreover, CQDs interact with uncoordinated lead ions in grain boundaries, effectively mitigating moisture-induced degradation. These results will advance the development of efficient perovskite solar cells (PSCs) for various renewable energy applications.

## Figures and Tables

**Figure 1 molecules-29-02556-f001:**
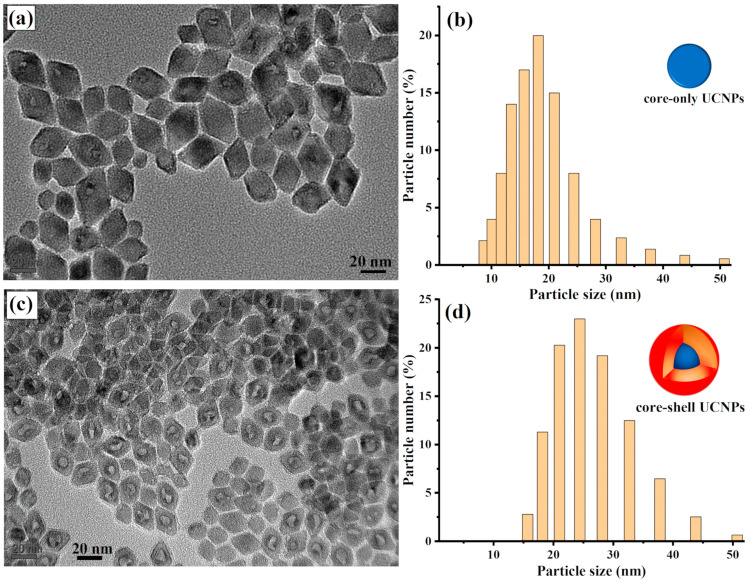
Characterizations of the synthesized UCNPs. (**a**) TEM analysis displays the core UCNPs, showing small, well-dispersed nanoparticles with an average size of 18 nm. (**b**) DLS measurement provides the size distribution of the core UCNPs, confirming the uniformity seen using TEM. The inset presents a systematic illustration of the UCNP core structure. (**c**) TEM of core–shell UCNPs reveals well-dispersed core–shell nanoparticles averaging 25 nm in size. (**d**) DLS confirmation validates the 25 nm average size of the core–shell UCNPs, confirming a consistent core–shell formation. The inset presents a systematic illustration of the UCNP core–shell structure.

**Figure 2 molecules-29-02556-f002:**
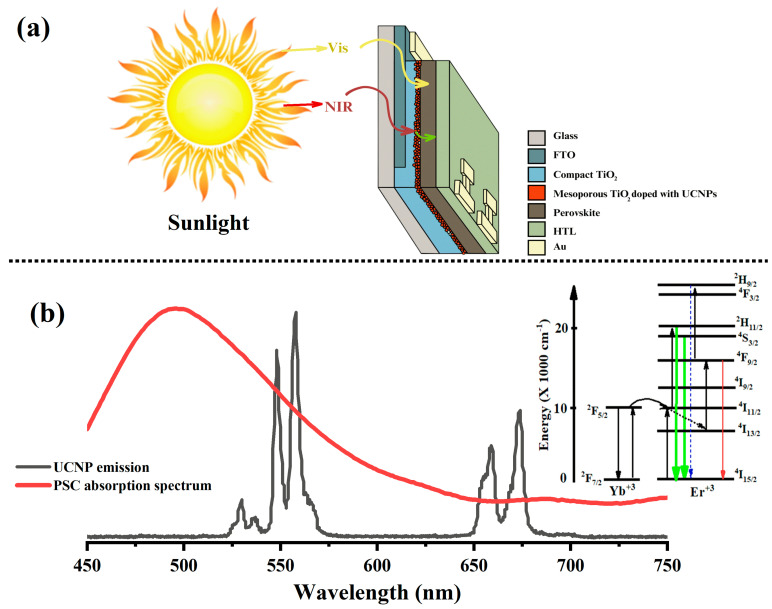
(**a**) Detailed schematic of the synthesized UCNPs integrated into perovskite solar cells (PSCs). The synthesized YLiF_4_:Yb,Er UCNPs absorb near-infrared (NIR) photons from sunlight and subsequently convert them into visible light. This conversion is crucial for enhancing the efficiency of the light-harvesting layer in the PSCs, enabling them to utilize a broader spectrum of solar radiation. (**b**) The overlap of the PSC absorption bands with UCNP emissions demonstrates how the absorption bands of the PSCs align with the emission peaks of the erbium-doped UCNPs (this UCNP emission was recorded from the core-only UCNPs (18 nm) and plotted for illustration purposes, while the optical emission from the core–shell UCNPs used in PSC fabrication will be shown later in this study). Specifically, the green emission peak at 550 nm and the red emission peak from 650 to around 680 nm from the UCNPs correspond closely with the spectral sensitivity regions of the perovskite layers in the solar cells. This alignment ensures that the light emitted by the UCNPs is effectively absorbed by the PSCs, optimizing the overall conversion of solar energy to electricity.

**Figure 3 molecules-29-02556-f003:**
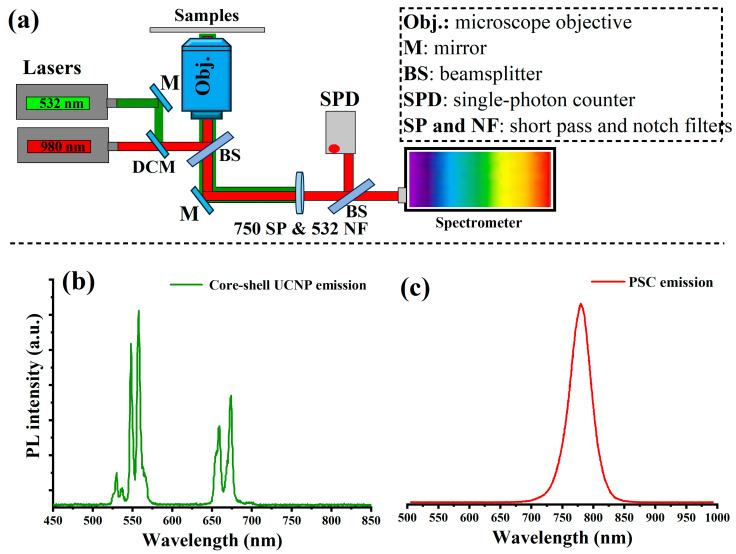
(**a**) Schematic illustration of a home-made confocal microscope designed and equipped with a green 532 nm laser and a near-infrared 980 nm laser for photoluminescence (PL) measurement of the PSC layers integrated with UCNPs in a core–shell structure. The designed optical microscope is equipped with an imaging system, lasers, a photon counter, and a custom-made spectrometer. (**b**) The UCNP emission spectrum measured directly from the core–shell UCNP layer (the particles that were integrated into the PSC fabrication), showing green emission peaks centered at 527 nm and 550 nm as well as a weak red emission peak at 650–680 nm. (**c**) The optical emission from the perovskite material under green (532 nm) excitation.

**Figure 4 molecules-29-02556-f004:**
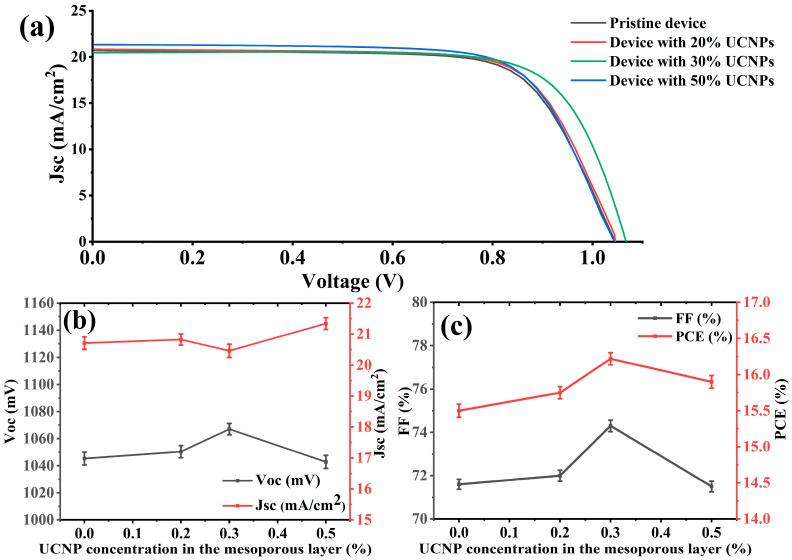
Performance parameters of the fabricated PSCs integrated with UCNPs in a core–shell structure. (**a**) presents the current–voltage (J-V) characteristic curves of fabricated PSCs measured under AM 1.5 G solar stimulation, comparing cells with varying amounts of UCNPs integrated into their mesoporous layers to those without UCNPs. The curves illustrate the impact of UCNPs on the electrical performance of the solar cells. (**b**,**c**) present key performance parameters as functions of the UCNP content. They display how the open-circuit voltage (Voc), short-circuit current density (Jsc), fill factor (FF), and power conversion efficiency (PCE) of the PSCs varied with different concentrations of UCNPs within their mesoporous layers.

**Figure 5 molecules-29-02556-f005:**
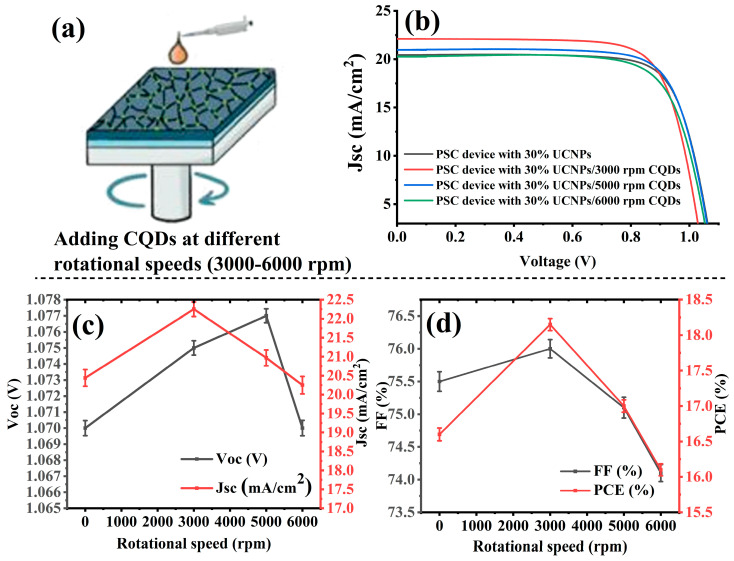
Performance parameters of the fabricated PSCs integrated with 30% UCNPs and CQDs at different spin coating speeds. (**a**) presents an illustration of adding CQDs on top of the perovskite layer of the fabricated PSCs at different spin coating speeds. (**b**) presents the current–voltage (J-V) characteristic curves of fabricated PSCs measured under AM 1.5 G solar stimulation, comparing cells with varying amounts of UCNPs integrated into their mesoporous layers to those without UCNPs. The curves illustrate the impact of UCNPs on the electrical performance of the solar cells. (**c**,**d**) present key performance parameters as functions of the UCNP content. They display how the open-circuit voltage (Voc), short-circuit current density (Jsc), fill factor (FF), and power conversion efficiency (PCE) of the PSCs varied with different concentrations of UCNPs within their mesoporous layers.

**Figure 6 molecules-29-02556-f006:**
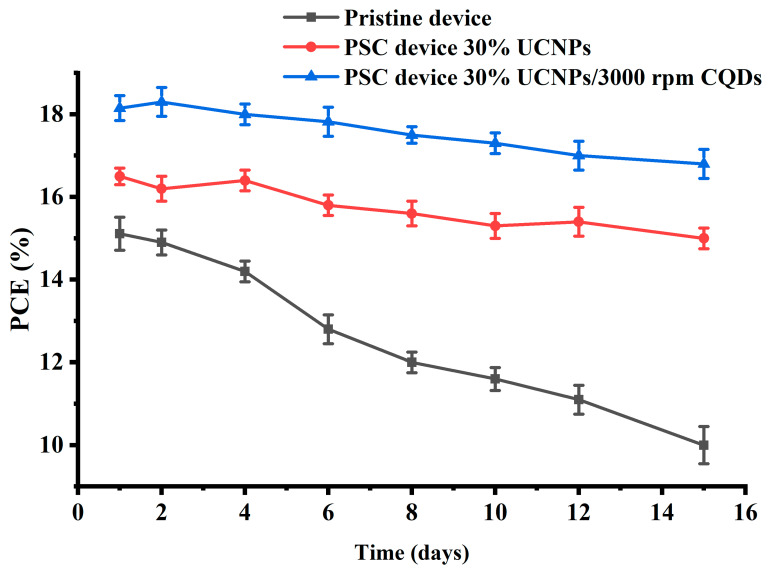
Power conversion efficiency (PCE) values of perovskite solar cell (PSC) devices incorporating upconversion nanoparticles (UCNPs) and carbon quantum dots (CQDs), compared to those without these additives. Measurements were taken over several days in ambient air, with no control over humidity levels. These data provide insight into the performance stability and degradation patterns of the PSC devices under real-world environmental conditions, highlighting the impacts of UCNPs and CQDs on device efficiency over time.

**Table 1 molecules-29-02556-t001:** Photovoltaic performances of the fabricated solar cells with different concentrations of UCNPs within the mesoporous layer.

Sample	Jsc (mA/cm^2^)	FF (%)	Voc (V)	PCE (%)
Pristine device	20.71 ± 0.15	71.6 ± 0.5	1.045 ± 0.007	15.5 ± 0.12
Device with 20% UCNPs	20.82 ± 0.18	72.0 ± 0.6	1.050 ± 0.008	15.75 ± 0.15
Device with 30% UCNPs	20.46 ± 0.20	74.3 ± 0.7	1.067 ± 0.009	16.22 ± 0.17
Device with 50% UCNPs	21.34 ± 0.19	71.5 ± 0.6	1.042 ± 0.008	15.90 ± 0.16

**Table 2 molecules-29-02556-t002:** Summary of the performance of the 30% UCNP-doped mesoporous TiO_2_ PSCs (reference device and CQD deposition at 3000–6000 rpm) measured under AM 1.5 G solar stimulation.

Sample	Jsc (mA/cm2)	FF (%)	Voc (V)	PCE (%)
PSC device 30% UCNPs	20.44 ± 0.18	75.5 ± 0.6	1.070 ± 0.009	16.65 ± 0.16
PSC device 30% UCNPs/3000 rpm CQDs	22.25 ± 0.21	76.0 ± 0.6	1.075 ± 0.010	18.15 ± 0.18
PSC device 30% UCNPs/5000 rpm CQDs	20.97 ± 0.19	75.1 ± 0.6	1.077 ± 0.009	16.96 ± 0.17
PSC device 30% UCNPs/6000 rpm CQDs	20.25 ± 0.17	74.1 ± 0.6	1.070 ± 0.009	16.06 ± 0.15

## Data Availability

The data presented in this study are available on request from the corresponding author.

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
