# Peer review of "Fabrication of Erbium-Doped Upconversion Nanoparticles and Carbon Quantum Dots for Efficient Perovskite Solar Cells"

_molecules, 2024, doi:10.3390/molecules29112556_

Round 1

Reviewer 1 Report

Comments and Suggestions for Authors
  1. While the paper outlines the synthesis of Erbium-doped upconversion nanoparticles (UCNPs) and carbon quantum dots (CQDs) and their application in high-efficiency perovskite solar cells, the experimental conditions for processes such as hot pressing (including temperatures, durations, and pressures) should be elaborated upon to enhance the reproducibility of the experiments.

  2. The study successfully demonstrates the improved performance of perovskite solar cells with the incorporation of UCNPs and CQDs; however, a more in-depth analysis of the synergistic mechanisms by which these materials enhance light absorption, reduce carrier recombination, and boost stability is warranted. The inclusion of theoretical models or computational simulations to support the observed phenomena is recommended.

  3. The mentioned efficiency improvements, like an increase in PCE from 16.22% to 18.15%, should be benchmarked against the current highest efficiencies achieved in perovskite solar cells, with explicit mention of the conditions under which these efficiencies were realized. Additionally, comparative performance assessments with control groups lacking doping and those using either UCNPs or CQDs alone would highlight the superiority of their combined usage.

  4. Although the positive influence of CQDs on environmental stability is acknowledged, more systematic long-term stability testing is necessary, encompassing degradation behavior under varied humidity, temperature, and light exposure conditions, to comprehensively assess their potential in practical applications.

  5. The reference list of the article should incorporate the latest research findings, especially recent advancements in the field of perovskite solar cells, to ensure the contemporaneity and relevancy of the study.

Comments on the Quality of English Language

Moderate editing of English language required.

Author Response

Dear Professor,

Best regards,

Masfer

Reviewer 2 Report

Comments and Suggestions for Authors

In this manuscript, UCNPs with down/up converting ion and CQDs were used for perovskite solar cells (PSCs). Compared to the pristine PSCs, the fabricated PSC device with 30% UCNPs passivated with CQDs at 3000 rpm spin coating speed showed improved PCE, from 16.65% to 18.15%, a higher photocurrent from 20.44 to 22.25 25 mA/cm2. The paper can be published with minor revisions as follows.

1. Using spin coating speed as parameter for PSC fabrication is difficult for readers to repeat the test. Can the author give the thickness or other parameters for the PSCs at different spin speeds?

2. I didn’t find the details for the preparation or properties of the CQDs used in this paper. Can the authors provide this information?

3. There are many typos or grammatical or style errors throughout the manuscript. The authors should check and revise them carefully. Here I mention some of them.

3.1 Lines 41-44: “GaAs solar cells, which were developed from second-generation solar cells, are extremely efficient technology it's efficiency has exceeded 30% but are far too costly for use in large-area terrestrial applications [5].” There is a grammatical error in this sentence.

3.2 See Line 81 and Line 91, the number should be used with the same style, such as 17.0% to 19.3%.

3.3 From Figure 2, the red emission should be at 650 to around 680 nm.

3.4 In Figure 3b, the X axis should be Wavelength (nm).

3.5 In “3. Materials and Methods” part, there are many typos or style errors, such as, “In two- neck flask”, “50 °C ØŒand”, “methanol ,4.0 mmol”, …

3.6 “Preparation of Ligand-Free” should be “Preparation of Ligand-Free UCNPs”.

3.7 “The substrates used to fabricate the PSCs, has a dimension of” should be “The substrates used to fabricate the PSCs have a dimension of”.

3.8 “We prepared an acidic ethanol with PH of 1 by adding 2.5 μL of hydrochloric acid to 40 334 mL of Ethanol. Then 1 mL of UCNPs were added to” should be “We prepared an acidic ethanol with pH of 1 by adding 2.5 μL of hydrochloric acid to 40 mL of ethanol. Then 1 mL of UCNPs was added to”.

Comments on the Quality of English Language

Moderate editing of English language required

Author Response

Dear Professor,

Best regards,

Masfer

Reviewer 3 Report

Comments and Suggestions for Authors

Reviewer comments:

In this work, the authors report on the synthesis of UCNPs doped with Erbium towards achieving UV and NIR light harvesting. The developed UCNPs were characterized by TEM, before being employed in PSC devices for improving performance. Indeed, their introduction within a mesoporous layer of the PSC device resulted to an improvement of photovoltaic parameters, when compared to the reference device without the UCNPs. Thus, demonstrating the validity of this approach towards improving performance. The paper is generally well written, however several points need to be addressed. Thus, in order to be considered for publication in Molecules (MDPI), the authors are strongly encouraged to address the points listed below (major revisions).

Suggested revisions and recommendations:

1. Page 2, lines 69-85: The authors refer to strategies of previous studies for improving performance and stability of PSCs. It would be good to mention the approach of laser-assisted crystallization of the perovskite active layer, in this literature paragraph (X. Huang et al., ACS Nano 14, 3150, 2020, and I. Konidakis et al., ACS Appl. Energy Mater. 1, 5101, 2018).

2. Page 4, Fig. 1: Each TEM image include two scale bars (one grey and one black) for 20 nm. While referring to 20 nm, the actual size of the bars is different within each figure (black appears larger). Please check and correct. Also, please explain in the caption what do the schematic insets of b and d depict?

3. Page 5, Fig. 2: The figure depicts the UCNPs emission spectrum. Would that be depended on the size of the so-formed particles (18 nm or 25 nm)? Which UPCNPs were employed in the actual PSC device. Please clarify this within the text and the figure caption.

4. Page 6, lines 174-176: It is stated that Fig. 3b presents the UCNPs emission. What is the difference between that spectrum and the one previously shown in Fig. 2b. Please clarify. Also, what is the difference between the emission spectrum shown in the inset of Fig. 3a and 3c? Please clarify. If there is no particular reason, please remove duplicate spectra in order to avoid confusion.

5. Page 7, Table 1: Please add error values in Table 1 (same applies for Table 2).

6. General comment: Did the authors perform any stability studies of the developed devices to check if the introduction of UCNPs improves stability apart from performance?

Author Response

Dear Professor,

Best regards,

Masfer

Round 2

Reviewer 1 Report

Comments and Suggestions for Authors

 Accept in present form.

Comments on the Quality of English Language

Minor editing of English language required.

Reviewer 3 Report

Comments and Suggestions for Authors

The author's have addressed all the points.